# Synthetic Post-Contrast Imaging through Artificial Intelligence: Clinical Applications of Virtual and Augmented Contrast Media

**DOI:** 10.3390/pharmaceutics14112378

**Published:** 2022-11-04

**Authors:** Luca Pasquini, Antonio Napolitano, Matteo Pignatelli, Emanuela Tagliente, Chiara Parrillo, Francesco Nasta, Andrea Romano, Alessandro Bozzao, Alberto Di Napoli

**Affiliations:** 1Neuroradiology Unit, Department of Radiology, Memorial Sloan Kettering Cancer Center, 1275 York Ave, New York, NY 10065, USA; 2Neuroradiology Unit, NESMOS Department, Sant’Andrea Hospital, La Sapienza University, Via di Grottarossa 1035, 00189 Rome, Italy; 3Medical Physics Department, Bambino Gesù Children’s Hospital, IRCCS, Piazza di Sant’Onofrio, 4, 00165 Rome, Italy; 4Radiology Department, Castelli Hospital, Via Nettunense Km 11.5, 00040 Ariccia, Italy; 5Neuroimaging Lab, IRCCS Fondazione Santa Lucia, 00179 Rome, Italy

**Keywords:** artificial intelligence, synthetic imaging, virtual contrast, augmented contrast, MRI, CT, gadolinium-based contrast agents, iodinated contrast agents, neuroimaging, cardiac imaging

## Abstract

Contrast media are widely diffused in biomedical imaging, due to their relevance in the diagnosis of numerous disorders. However, the risk of adverse reactions, the concern of potential damage to sensitive organs, and the recently described brain deposition of gadolinium salts, limit the use of contrast media in clinical practice. In recent years, the application of artificial intelligence (AI) techniques to biomedical imaging has led to the development of ‘virtual’ and ‘augmented’ contrasts. The idea behind these applications is to generate synthetic post-contrast images through AI computational modeling starting from the information available on other images acquired during the same scan. In these AI models, non-contrast images (virtual contrast) or low-dose post-contrast images (augmented contrast) are used as input data to generate synthetic post-contrast images, which are often undistinguishable from the native ones. In this review, we discuss the most recent advances of AI applications to biomedical imaging relative to synthetic contrast media.

## 1. Introduction

Contrast media are essential tools in biomedical imaging, allowing for more precise diagnosis of many conditions. The main contrast agents employed in radiology are iodinated contrasts for CT imaging and gadolinium-based contrast agents (GBCA) for MRI. As for any molecule used for medical purposes, contrast media are not exempt from contraindications and side-effects, which need to be evaluated against the known diagnostic benefits when scans are ordered in clinical practice.

One of the main side-effects of iodinated contrasts, besides allergic reactions, is nephrotoxicity [1]. Iodinated contrast media may lead to acute kidney injury (AKI) in certain patients, and they represent a leading cause of hospitalization [1]. Although the exact mechanism of renal damage is still debated, there is evidence of direct cytotoxicity of iodinated contrasts on tubular epithelial and endothelial linings of the kidney. Additionally, these contrasts seem to affect the renal hemodynamics, due to increased oxygen radicals synthesis and reduction in blood flow in both glomerular and tubular capillaries related to hyperviscosity [1]. GBCA toxicity was first reported as a multi-systemic condition known as nephrogenic systemic fibrosis (NSF), described in few cases and mainly related to renal failure [2]. More recently, brain GBCA deposition was discovered through imaging [3,4,5], animal models [6], and autopsy studies [7,8]. Brain accumulation of GBCAs depends on their chemical structure, with higher de-chelation susceptibility for linear compounds. Such evidence led to the suspension of linear GBCA from commerce [9], leading to changes in clinical guidelines for contrast administration [10]. The existence of a definite clinical correlate for GBCA brain deposition is still debated; however, different types of toxicity may occur in the body according to the patient’s age and clinical state [3]. Oxidative stress may play a role in gadolinium ions toxicity, as reflected by changes in intracellular glutathione levels [11,12]. In this situation, finding new means to boost biomedical imaging diagnostic power with low-dose contrast administration, and finding possible alternative diagnostic methods to contrast media for common disorders [13], are extremely relevant in current clinical practice.

In the last few years, artificial intelligence (AI) has revolutionized the field of medicine, with remarkable applications in biomedical imaging [14]. Due to the high volume of imaging data stored in PACS archives, AI applications have proven capable of fueling predictive models for differential diagnosis and patient outcome in many different conditions [14,15,16,17,18,19]. Most recently, the field of ‘virtual’ and ‘augmented’ contrast emerged from the intersection of AI and biomedical imaging. The idea behind these applications is to create virtual enhancement starting from available information on non-contrast images acquired during the same scan (virtual contrast) or to augment the enhancement obtained from a low-dose administration (augmented contrast), through AI computational modeling.

In this review, we present the most recent advances in AI applications to biomedical imaging relative to contrast media. This paper is divided into three sections: (1) we discuss the main AI methods currently available to generate virtual and augmented contrast; (2) we review the main applications of AI-powered contrast media to neuroradiology; and (3) we review the main applications of AI-powered contrast media to body imaging. Finally, we present our reflections on possible future developments.

## 2. AI Architectures in Synthetic Reconstruction

The term AI was formally proposed for the first time in 1956 at a conference at Dartmouth University, to describe the science of simulating human intellect behaviors through computers, including learning, judgment, and decision making [20]. The application of AI in medicine relies on machine learning and deep learning algorithms. These powerful mathematical algorithms can discover complex structure in high-dimensional data by improving learning through experience. There are three main types of algorithm learning: (i) unsupervised, which is the ability to find patterns in data without a priori information; (ii) supervised, which is used for data classification and prediction based on ground truth; and (iii) reinforcement learning, a technique that enables learning by the trial-and-error approach [21]. Among the many applications of AI in medicine, recent years have witnessed a rising interest towards medical image analysis (MIA). In this context, deep learning networks can be applied to solve classification problems, and for segmentation, object detection, and synthetic reconstruction of medical images [22].

Virtual and augmented contrasts can be considered an application of AI in the field of synthetic imaging. Convolutional neural networks (CNNs) and generative adversarial networks (GANs) are the only known deep learning tools used for image reconstruction [23,24], due to their ability to capture image features that describe a high level of semantic information. These two groups of machine learning architectures have achieved considerable success in the MIA field, and they are further explored in the following sections with particular attention on deep learning architectures to synthesize new images (synthetic post-contrast images) from pre-existing ones (either non-contrast images or low-dose post contrast images) [25]. Previous studies relevant to this topic are summarized in Table 1.

### 2.1. Convolutional Neural Networks in Synthetic Reconstruction

A CNN can be considered as a simplified version of the neocognitron model introduced by Fukushima [26] in 1980 to simulate the human visual system. CNNs are characterized by an input layer, an output layer, and multiple hidden layers (i.e., convolutional layer, pooling layer, fully connected layer, and various normalization layers). Due to their architecture, the typical use of CNNs is for image classification tasks, where the networks’ output is a single class label related to an input image [27]. When addressing CNNs’ use in medical research, the main applications here are also targeted toward classification problems [28,29]. However, the current literature includes several examples of successful implementation of CNN architectures to address other machine learning and computer vision challenges, such as image reconstruction and generation.

Gong et al., implemented a workflow with zero-dose pre-contrast MRIs and 10% low-dose postcontrast MRIs as inputs to generate full-dose MRI images through AI (augmented contrast) [30]. In this study, the model consisted of an encoder-decoder convolutional neural network (CNN) with three encoder steps and three decoder steps based on U-net architecture. The basic architecture of U-Net consists of a contracting part to capture features, and a symmetric expanding part to enable precise localization and reconstruction. A slightly modified version of the same model was tested by Pasumarthi et al., with similar results [31]. Xie et al., also investigated a U-net based approach to obtain contrast-enhanced MRI images from non-contrast MRI [32]. This type of network architecture has demonstrated superior performance in medical imaging tasks such as segmentation and MRI reconstruction [33,34,35].

### 2.2. Generative Adversarial Networks in Synthetic Reconstruction

Another approach to produce synthetic post-contrast images relies on generative adversarial networks (GANs), which can be used to generate images from scratch, showing increasing applications in current literature. Starting from a set of paired data, these networks are able to generate new data that are indistinguishable from the ground truth. Current state-of-the-art applications of GANs include image denoising and quality improvement algorithms, such as Wasserstein GAN (WGAN) [36] and the deep convolutional GAN (DCGAN) [37,38,39,40,41,42,43,44]. Regarding image segmentation and classification, another extension of the GAN architecture—the conditional GAN (c-GAN) demonstrated promising results [45]. Additionally, Pix2Pix GANs [46] are a general GAN approach for image-to-image translation, which is very useful for generating new synthetic MRI images starting from images of a different domain.

The most used GAN architectures to obtain synthetic diagnostic images ex novo from MRI and CT scans are the CycleGAN networks [37,43,47]. These are a class of deep learning algorithms that rely on the simultaneous training of two networks: the first is focused on data generation (alias generator) and the second is focused on data discrimination (alias discriminator). The two neural networks compete against each other, learning the statistical distribution of the training data, which allows the generation of new examples from the same distribution [48]. Specifically, the two generator models and the two discriminator models mentioned above are simultaneously trained. One generator takes images from the first domain and generates images for the second domain. Conversely, another generator takes images from the second domain and outputs images for the first domain. The discriminator models determine the plausibility of the generated images. An idea of cycle consistency is used in the CycleGAN. This means that the output image of the second generator should match the original image if the image output by the first generator was used as input to the second generator. A schematic picture of the described architecture can be seen in Figure 1. In all the reported studies, these networks are capable of generating true diagnostic images from a qualitative and informative point of view, demonstrating the enormous advantages of this type of application.

### 2.3. Model Implementation

Model implementation typically includes three steps: (i) training: the model is trained on an appropriate dataset for the given task; (ii) internal validation: during training, the model is continuously validated on data that is not part of the training set to evaluate the model’s performance; and (iii) external validation: after training, the model is tested on a separate dataset, from which the final metrics are extrapolated [49]. Model internal and external validation are extremely important steps to understand its robustness and reliability. Internal validation involves collecting and analyzing information about the model’s characteristics and outcomes, with the main purpose to ensure that the model is operating correctly. External validation proves the generalizability of the model, which is a prerequisite for clinical use [50]. Steps (ii) and (iii) are key operations to assess the efficacy of a model during both initial research and development phases.

The choice of which metric to use, in steps (ii) and (iii), is strictly related to the type of model data input and output to be compared [51]. For the purposes of generating synthetic contrast, it is expected that the reconstructed images maintain the same information content and quality as the original images to guarantee a correct diagnosis by medical physicians. Bustamante et al., carried out a visual inspection of the synthetic images obtained with GANs, finding no major differences compared to real images [24]. In experimental settings, different scores can be used to test the equivalence of generated vs. native images. An example of a rating score is the mean opinion score (*MOS*), an index score of quality ranging between 1 and 5 (1 as the most inferior value) [52]:(1)$$MOS=\frac{\sum_{n=1}^{N}R_{n}}{N}$$

*R*: individual ratings for the object given by *N* subjects.

As a quantitative evaluation, the current literature proposes the structure similarity index (*SSIM*), which accounts for luminance, contrast, and degradation of structural information between two images [33]. *SSIM* principally refers to a structural information variation, providing a good approximation of distortion in comparing images. Where *x* and *y* the two nonnegative image signals, the measure is expressed as follows:(2)$$SSIM\;\left(x,y\right)=\frac{\left(2\mu_{x}\mu_{y}+c_{1}\right)\left(2\sigma_{xy}+c_{2}\right)}{\left(\mu_{x}{}^{2}+\mu_{y}{}^{2}+c_{1}\right)\left(\sigma_{x}{}^{2}+\sigma_{y}{}^{2}+c_{2}\right)}$$
$\mu_{x,\;}\sigma_{x}{}^{2}$: average and variance of *x*$\mu_{y},\;\sigma_{y}{}^{2}$: average and variance of *y*$\sigma_{xy}$: covariance$c_{1}$, $c_{2}$: two variables to stabilize the division with weak denominator


Further information about this metric can be found in the study of Wang et al. [53]

Another commonly used measure to quantify image reconstruction quality is the peak signal-to-noise ratio (*PSNR*) metric. This is evaluated as the ratio between the maximum possible power of an image and the power of corrupting noise that affects its quality [34]. In other words, to estimate the *PSNR* of an image, it is necessary to compare the real image to an ideal one with the maximum possible power.
(3)$$PSNR=10\;\log_{10}\left(\frac{MAX_{I}^{2}}{MSE}\right)$$
$MAX_{I}$: maximum possible pixel value of the image*MSE*: mean square error


As per its definition, higher *PSNR* values equal better quality of the reconstructed output images [54].

**Table 1 pharmaceutics-14-02378-t001:** Synthetic reconstruction studies, including AI architectures, database used, specific algorithms, and task characteristics. In particular, the database used is expressed according to not-specified, private, or public availability, with the total number of patients included in brackets (N); when the database is public, the name is reported. CNN = convolutional neural network; GAN = generative adversarial network; BraTS = Brain Tumor Segmentation database.

AI Architecture	Reference	Database (N)	Specific Algorithm	Task
CNN	[30]	Private (60)	U-net based	To obtain 100% full-dose 3D T1-weighted images from 10% low-dose 3D T1-weighted images
	[31]	Private (640)	U-net based	To obtain 100% full-dose 3D T1-weighted images from 10% low-dose 3D T1-weighted images and pre-contrast 3D T1-weighted images
	[32]	Brats2020 (369)	U-net based	To obtain contrast-enhanced T1-weighted images from non-contrast-enhanced T1-weighted images
GAN	[37]	Not specified (222)	CycleGAN	To obtain high blood-tissue contrast from non-contrast 4D flow MRI
	[43]	Private (26)	2D-CycleGAN	To obtain contrast-enhanced CT from non-contrast-enhanced CT
	[46]	Private (48)	Conditional GAN	To obtain fat saturated T1-weighted images from non-contrast-enhanced T1-weighted images
	[47]	Brats2015 BraTS2015 (50)	CycleGAN with Attention algorithm integrated	To obtain contrast-enhanced T1-weighted images from non-contrast-enhanced T1-weighted images

## 3. Clinical Applications in Neuroradiology

In neuroradiology, the contrast enhancement of a lesion inside the brain tissue reflects a blood–brain barrier rupture. This event occurs in many different brain diseases such as some primary tumors and metastases, neuroinflammatory diseases, infections, and subacute ischemia, for which contrast injection during MRI examination is considered mandatory for accurate assessment, differential diagnosis, and monitoring [55,56,57,58,59,60]. Other uses of contrast include the evaluation of vascular malformation and aneurysms. In this scenario, the implementation of AI algorithms to reduce contrast usage could result in significant benefit for the patients, and reduced scan times and costs [61]. However, the main drawback of AI analysis, especially through deep learning (DL) methods, reside in the need for a large quantity of data. For this reason, the literature of AI virtual contrast in neuroradiology is focused on MRI examination of relatively common diseases such as tumors and multiple sclerosis (MS) (Table 2).

### 3.1. AI in Neuro-Oncology Imaging

In neuro-oncology, contrast enhancement is particularly useful not only as a marker for differential diagnosis and progression, but it is also considered the target for neurosurgical removal of a lesion and an indicator of possible recurrence. Although in recent years some authors suggested expanding the surgical resection of brain tumors beyond the contrast enhancement [62,63], injection of gadolinium remains a standard for both first and follow-up MR scans. To avoid the use of gadolinium, Kleesiek et al., applied a Bayesan DL architecture to predict contrast enhancement from non-contrast MR images in patients with gliomas of different grades and healthy controls [64]. The authors obtained good results in terms of qualitative and quantitative assessment (approximate sensitivity and specificity of 91%). Similarly, other studies applied a DL method to pre-contrast MR images of a large group of glioblastomas and low-grade gliomas with a good structural similarity index between simulated and real postcontrast imaging, and the ability of the neuroradiologist to determine the tumor grade [32,65]. These methods can also be applied to sequences different from the T1. Recently, Wang et al., developed a GAN to synthetize 3D isotropic contrast-enhanced FLAIR images from a 2D non-contrast FLAIR image stack in 185 patients with different brain tumors [66]. Interestingly, the authors went beyond simple contrast synthesis and added super-resolution and anti-aliasing tasks in order to solve MR artifacts and create isotropic 3D images, which give a better visualization of the tumor, with a good structural-similarity index to the source images [66]. Calabrese et al., obtained good results in synthetize contrast-enhanced T1 images from non-contrast ones in 400 patients with glioblastoma and lower-grade glioma [65]. In addition, the authors included an external validation analysis, which is always recommended in DL-based studies [65]. However, the simulated images appeared blurrier than real ones, a problem that could especially affect discriminating progression in follow-up exams [65]. This shortcoming appears to be a common issue of all ‘simulated imaging’ studies. As stated above, contrast enhancement reflects disruption of the blood–brain barrier, information that is usually inferred from the pharmacokinetics of gadolinium-based contrasts within the brain vasculature; this explains the difficulty of generating information from sequences that may not contain it. Moreover, virtual contrast may hinder interpretation of derived measures, such as perfusion-weighted imaging, which have been proven crucial for differential diagnosis and prognosis prediction in neuro-oncology [15,16,67,68]. Future directions could make use of ultrahigh field scanners that may have enough resolution to be closer to molecular imaging. In the meantime, another approach has been explored to address the issue. Rather than eliminating contrast injection, different studies used AI algorithms to enhance a reduced dose of gadolinium (10% or 25% of the standard dose), a method defined as ‘augmented contrast’ [30,31,69,70]. This method, used on images of different brain lesions, including meningiomas and arteriovenous malformations, allows the detection of a rupture of the blood–brain barrier with a significantly lower contrast dose. Another advantage is the better quality of the synthetized images, as perceived by evaluating neuroradiologists [30,31]. Such benefits persist with data obtained across different scanners, including both 1.5T and 3T field strengths, a fundamental step for the generalizability of results [30,69,70]. Nevertheless, augmented contrast techniques are not exempt from limitations. Frequently encountered issues with these techniques are the difficulty of detecting small lesions, the presence of false positive enhancement, probably due to flow or motion artifacts, and coregistration mismatch [31,70]. Another concern is the lack of time control between the two contrast injections. Most of the studies perform MRI in a single session by acquiring the low-dose sequence first, followed by full-dose imaging after injecting the remaining dose of gadolinium [30,31,69,70]. The resulting full-dose images are, thus, a combination of the late-delayed pre-dose (10% or 25%) and the standard-delayed contrast, which can result in a slightly different enhancement pattern from the standard full-contrast injection. Future studies could acquire low and standard dose exams on separate days, with controlled postcontrast timing. Lastly, future directions could include prediction of different contrast imaging. In fact, other types of contrast are being developed to give additional information on pathologic processes, such as neuroinflammation [71]. Another interesting application of AI in neuro-oncology imaging consists of augmenting the contrast signal ratio in standard dose T1 images, in order to better delineate tumors and detect smaller lesions. Increasing contrast signal for better detection of tumors, in fact, has always been a goal when developing new MR sequences, leading recently to a consensus for brain tumor imaging, especially for metastasis [72]. A recent study by Ye et al., used a multidimensional integration method to increase the signal-to-noise ratio in T1 gradient echo images [73], also resulting in contrast signal enhancement. By comparison, Bône et al., implemented an AI-based method to increase contrast signal similarly to the ‘augmented contrast’ studies [74]. The authors trained an artificial neural network with T1 pre-contrast, FLAIR, ADC, and low-dose T1 (0.025 mmol/kg). Once trained, the model was leveraged to amplify the contrast on routine full-dose T1 by processing it into high-contrast T1 images. The hypothesis behind this process was that the neural network learned to amplify the difference in contrast between pre-contrast and low-dose images. Hence, by replacing the low-dose sequence with a full-dose one, the synthesis led to a quadruple-dose contrast [74]. The results led to a significant improvement in image quality, contrast level, and lesion detection performance, with a sensitivity increase (16%) and similar false detection rate with respect to routine acquisition.

### 3.2. AI in Multiple Sclerosis Imaging

MS is the most common demyelinating chronic disorder, and mostly affects young adults [75]. MRI is a fundamental tool in both MS diagnosis and follow-up, and gadolinium injection is usually mandatory. In fact, contrast enhancement is essential to establish dissemination in time according to the McDonald’s criteria, and to evaluate ‘active’ lesions in follow-up scans [76]. Due to the relatively young age at diagnosis, MS patients undergo numerous MRIs throughout the years. For this reason, different research groups are trying to limit contrast administration in selected cases to minimize exposition and costs [40,53]. One study tried to develop a DL algorithm to predict enhancing MS lesions from non-enhanced MR sequences in nearly 2000 scans, reporting accuracy between 70 and 75%, leading the way to further research [77]. The authors used only conventional imaging such as T1, T2, and FLAIR. Future studies could add DWI to the analysis as this sequence has been proven to be useful in identifying active lesions [78]. Small dimensions of MS plaques, as already seen above for brain tumor studies, could be a reason for the low accuracy of synthetic contrast imaging; future directions could try enhancing low-dose contrast injections by means of AI algorithms.

In conclusion, virtual and augmented contrast imaging have an interesting role in neuroradiology for assessing different diseases for which gadolinium injection is, for now, mandatory.

**Table 2 pharmaceutics-14-02378-t002:** Studies in neuroradiology with field of application, database used, tasks, and results characteristics. In particular, the database used is expressed according to not-specified, private, or public availability with the total number of patients included in brackets (N); when the database is public, the name is reported. BraTs= Brain Tumor Segmentation database; SSIM = structural similarity index measures; PSNR = peak signal-to-noise ratio; FLAIR = fluid attenuated inversion recovery; PCC = Pearson correlation coefficient; SNR = signal-to-noise ratio; MS = multiple sclerosis; MR = magnetic resonance.

Field of Application	Reference	Database (N)	Task	Results
Neuro-oncology	[30]	Private (60)	To obtain 100% full-dose 3D T1-weighted images from 10% low-dose 3D T1-weighted images	SSIM and PNSR increase by 11% and 5 dB
	[31]	Private (640)	To obtain 100% full-dose 3D T1-weighted images from 10% low-dose 3D T1-weighted images and pre-contrast 3D T1-weighted images	SSIM 0.92 ± 0.02; PSNR 35.07 dB ± 3.84
	[32]	BraTs2020 (369)	To obtain contrast-enhanced T1-weighted images from non-contrast-enhanced MR images	SSIM and PCC 0.991 dB ± 0.007 and 0.995 ± 0.006 (whole brain); 0.993 ± 0.008 and 0.999 ± 0.003
	[47]	Private 1 (185) Private 2 (36) BraTs2018 (73 for SR)	To obtain 3D isotropic contrast-enhanced T2 FLAIR from non-contrast enhanced 2D FLAIR and image super resolution	SSIM 0.932 (whole brain), 0.851 (tumor region); PSNR 31.25 dB (whole brain) 24.93 dB (tumor region)
	[64]	Private (116)	To obtain contrast-enhanced T1-weighted images from non-contrast-enhanced MR images	Sensitivity 91.8%, Specificity 91.2%
	[65]	Private (400) BraTs2020 (286) (external validation)	To obtain contrast-enhanced T1-weighted images from non-contrast-enhanced MR images	SSIM 0.84 ± 0.05
	[69]	Private (83)	To obtain 100% full-dose 3D T1-weighted images from 10% low-dose 3D T1-weighted images	Image quality 0.73; image SNR 0.63; lesion conspicuity 0.89; lesion enhancement 0.87
	[70]	Private (145)	To obtain 100% full-dose 3D T1-weighted images from 25% low-dose 3D T1-weighted images	SSIM 0.871 ± 0.04; PSNR 31.6 dB ± 2
	[74]	Private (250)	To maximize contrast in full-dose 3D T1-weighted images	Sensitivity in lesion detection increase by 16%
Multiple Sclerosis	[77]	Private (1970)	To identify MS enhancing lesion from non-enhanced MR images	Sensitivity and specificity 78% ± 4.3 and 73% ± 2.7 (slice-wise); 72% ± 9 and 70% ± 6.3

## 4. Clinical Applications in Body Imaging

### 4.1. Abdominal and Thoracic Radiology

Imaging is a fundamental tool in both abdominal and thoracic imaging, ranging from diagnosis in the emergency setting to more complex tumor differential diagnosis, therapeutic planning, and follow-up. Contrary to neuroradiology, the implementation of synthetic contrast algorithms is hindered by two main issues: (1) respiration of the patients, which may lead to misalignment between different acquisitions, one of the main obstacles to the development of synthetic imaging [30]; (2) the presence of multiple different organs in these compartments, leading to possible anatomy misclassification. The latter is mainly a problem of abdominal imaging. However, some studies attempted to generate synthetic contrast imaging in these two contexts (Table 3).

MRI of liver neoplasms heavily relies on GBCA injection both for discerning liver cancer from other entities (i.e., hemangiomas) and because some tissues are invisible before GBCA injection [79]. A study by Zhao et al., used a tripartite-GAN model to generate contrast T1 images from non-contrast ones in 265 patients with either hepatocellular carcinoma or hemangioma, and a few subjects without lesions with a good signal-to-noise ratio and high tumor detection accuracy [80]. However, the results are rough enhanced images that rely mainly on tumor structural information, leaving out other important features such as presence of capsules and infiltrative growth. To address these shortcomings, Xu et al., developed a reinforcement learning model that relies on a pixel-level graph to evaluate the images and synthetize contrast-enhanced sequences [81], on 325 subjects with both benign and malignant tumors and healthy controls. The authors achieved a structural similarity index of 0.85 between synthetized and acquired images, prompting more feasible clinical use. However, both studies used only non-contrast T1 images for AI training. Future research should investigate the possibility of adding other sequences (T2, DWI) to boost model performance.

CT scans for abdominal pain are very common in the emergency room setting and very useful for diagnosis and decision making [82,83]. Since contrast injection is not required for all patients with abdominal pain, Kim et al., sought to increase non-contrast CT (NCCT) diagnostic performance by generating contrast-enhanced CT (CECT) through a DL algorithm in more than 500 patients (divided into training, test, and external validation sets) [84]. The consultant and in-training radiologists involved in the research reported increased confidence in diagnosing, especially oncologic conditions such as biliary disease or inflammatory conditions (appendicitis, pancreatitis, diverticulitis) [84]. However, the main drawback of the study was the increased confidence in both correct and incorrect diagnoses, raising some concerns about the utility of this approach. Future directions should focus more on diagnostic accuracy.

Contrast agents are not usually required for detection of lung parenchymal lesions on CT. Nonetheless, CECT is useful for the evaluation of other thoracic structures, such as mediastinum, pleura, and vessels. In this regard, Choi et al., generated CECT from NCCT in a small group of 63 patients (with external validation) acquired on scanners by different vendors and with various scanning parameters [85]. The authors evaluated the conspicuity of mediastinal lymph nodes, which was found to be higher on the synthetized images [85]. However, the AI was fed with ‘virtual no-contrast’ images obtained on dual-energy CT, which enabled perfect spatial registration between the input and the ground truth; nonetheless, it also raises concerns about real ‘contrast synthesis’ since obtaining virtual no-contrast imaging also requires contrast injection. Other interesting thorax CT applications include contrast synthesis for cardiac left chamber evaluation [86] and to better delineate the heart during radiotherapy for breast cancer [87], thus resulting in lower risk of cardiac radio-induced toxicity.

### 4.2. Cardiovascular Radiology

Computed tomography angiography (CTA) is a computed tomography technique that relies on iodine-based contrast administration to visualize arteries, and to diagnose and evaluate the related diseases, such aneurysms, dissection, or stenosis. With the widespread availability of state-of-the-art multidetector technology, CTA has become the imaging test of choice for various aortic conditions because of its excellent spatial resolution and rapid image acquisition. CTA provides a robust tool for planning aortic interventions and diagnosing acute and chronic vascular diseases. CTA is the standard for imaging aneurysms before intervention and evaluating the aorta in the acute setting to assess traumatic injury, dissection, and aneurysm rupture [88,89]. Furthermore, the recently published results of the DISCHARGE trial [90] support the use of CTA instead of invasive angiography for the assessment of coronary artery disease in patients with stable chest pain. For all these reasons, the diffusion of CTA will likely increment exponentially in the near future.

To avoid use of contrast, Chandrashekar et al., hypothesized that raw data, acquired from a NCCT, could be used to differentiate blood and other soft tissues [91]. Blood is predominantly fluid, with red/white blood cells, whereas a possible adjacent intraluminal thrombus (ILT) is predominantly fibrinous and collagenous, with red cells/platelets. These individual components vary in ultrastructure and physical density, which should be reflected in different (albeit subtle) Hounsfield Units (HUs) on a CT scan (either in individual HU values or in their spatial distribution/‘texture’). Using deep learning (DL) generative methods, these subtle differences can be amplified to enable simulation of contrast-enhanced images without the use of contrast agents.

In this study, the authors investigated the ability of generative adversarial networks (GANs) for this non-contrast to contrast image transformation task [92,93]. They first investigated differences between visually indistinguishable regions (lumen, interface, thrombus) within NCCT, comparing the HU intensity distribution and radiomic features. The latter have been used to find disease features that fail to be appreciated visually. Then, the authors showed that generative models enable the visualization of aortic aneurysm morphology in CT scans obtained without intravenous (IV) contrast administration, and that transformation accuracy was independent of aortic abdominal aneurysm (AAA) size and shape.

Some authors showed that the evolution of 3D indices, especially thrombus volume, is linked to AAA progression rupture risk, and even the incidence of adverse cardiovascular events [94,95,96]. Furthermore, assessing ILT spatial morphology is important for surgical planning and has been shown to influence postoperative outcomes, such as in the case of type 2 endoleak onset [97]. Hence, this evidence reinforces the clinical impact of using DL generative networks for image transformation. Using DL for pseudo-contrast CT visualization of AAA, ILT and its side branches from an NCCT is a promising technique and a safer alternative to the routinely obtained contrast-enhanced CTA. Future studies are needed to validate the clinical utility of such techniques, especially in pre-operative endovascular graft planning imaging.

In cardiac magnetic resonance (CMR), the use of GBCAs is essential for detecting focal myocardial lesions and fibrosis in a variety of cardiovascular diseases using late gadolinium enhancement (LGE) sequences [98,99]. The presence and extent of LGE are independent risk factors for adverse outcomes [100,101,102,103,104,105]. Conventional LGE is dependent on intravenous administration of GBCA and requires at least 10 min after injection to visualize contrast redistribution [106]. LGE image quality is dependent on appropriate adjustment of TI, although the phase-sensitive inversion recovery technique is less sensitive to the TI setting.

Zhang et al., hypothesized that native T1 maps may be transformed into images similar to LGE images [107]. Using novel AI approaches, a virtual native enhancement (VNE) imaging technology was developed [43,47], which exploits and enhances existing contrast and signals within the native T1 maps and cine frames, and displays them in a standardized presentation. The VNE imaging was then validated by comparison against matching LGE for image quality, visuospatial agreement, and myocardial lesion quantification. This approach was developed first in hypertrophic cardiomyopathy (HCM) because its features of regional heterogeneity and diverse tissue remodeling processes make it an ideal test case for a wide range of cardiac diseases.

The proposed VNE technology uses two components: native T1 mapping images, which provide image contrast and signal changes in myocardial tissue [108,109,110], and pre-contrast cine frames of a cardiac cycle for additional wall motion information and more defined myocardial borders. These images were fed to a DL generator to derive a VNE image: an AI technology transformed native T1 map (together with cines) into more readable presentation of LGE, ready for standard clinical interpretation [93,111,112,113]. The DL is effectively acting as a ‘virtual contrast agent’ creating ‘virtual LGE’ image from native CMR sequences. Zhang et al., showed that VNE images had significantly better quality than LGE images, demonstrating high agreement with LGE in myocardial lesion visuospatial distribution and quantification [107]. The clinical utility of detecting subtle lesions (often also seen in LGE) remains unclear, in line with literature reporting sensitivity of T1 to early myocardial changes in HCM patients.

Finally, the VNE technology has the potential to significantly improve clinical practice in CMR imaging, as it may allow significantly faster, lower-cost, and contrast-free CMR scans, enabling frequent monitoring of myocardial tissue changes.

### 4.3. Head and Neck Radiology

In head and neck radiology, MRI is a very useful diagnostic tool for identification, staging, radiotherapy and surgery planning, and treatment evaluation of certain malignancies, such as nasopharyngeal carcinoma (NPC) [114], especially at a stage when the disease is not visible on endoscopy [115]. GBCAs are used for their power to enhance detectability and boundaries of these tumors [116].

Very few applications of AI-powered contrast media have been proposed in head and neck imaging to date. An exploratory study fed a DL model with non-enhanced conventional MR sequences (T1- and T2-weighted), with the aim of distinguishing between NPC and benign hyperplasia, as well as assessing the T stage, in more than 4000 subjects. The model succeeded in discriminating NPC from benign hyperplasia with an accuracy comparable to a model that included T1 post-contrast sequences (99%) [117]. In addition, when both T1 and T2 non-contrast sequences were evaluated together, the model could predict T stage comparably to enhanced sequences [117].

Although more research is needed to confirm these results, including the need of external validation for generalizability of results, it represents a good example of contrast substitution in oncologic radiology, suggesting a test for other malignancies.

**Table 3 pharmaceutics-14-02378-t003:** Studies in body imaging with field of application, database used, tasks, and results characteristics. In particular, the database used is expressed according to not-specified, private, or public availability with the total number of patients included in brackets (N); when the database is public, the name is reported. SSIM = structural similarity index measure; PSNR = peak signal-to-noise ratio; PCC = Pearson correlation coefficient; MR = magnetic resonance; CT = computed tomography; ASD = average surface distance; DSC = dice similarity coefficient.

Field of Application	Reference	Database (N)	Task	Results
Abdominal Imaging	[80]	Private (265)	To obtain contrast-enhanced T1-weighted FS images from non-contrast-enhanced T1-weighted FS images	PNSR 28.8; accuracy 89.4%
	[81]	Private (325)	To obtain contrast-enhanced T1-weighted images from non-contrast-enhanced T1-weighted images	SSIM 0.85 ± 0.06; PCC 0.92
	[84]	Private (500)	To obtain contrast-enhanced CT from non-contrast-enhanced CT	Increased diagnostic confidence and accuracy of radiologist
Thoracic Imaging	[85]	Private (63)	To obtain contrast-enhanced CT from non-contrast-enhanced CT	SSIM 0.84; PNSR 17.44; contrast-to-noise ratio (lymph nodes) 6.15 ± 5.18
Head and Neck Imaging	[117]	Private (4478)	To distinguish NPC from hyperplasia in non-contrast-enhanced MR images only and including contrast-enhanced MR images in the model	Similar ASD and DSC between the two models

## 5. Conclusions and Future Directions

AI-powered contrast media have been proven to be capable of delivering impressive results in several conditions. While current applications to new diagnoses of enhancing lesions, especially in neuro-oncology [70], are still limited and need to be confirmed in larger studies, virtual and augmented contrasts demonstrated promising results in disease follow-up over time. This is especially true for multiple sclerosis, where the administration of contrast can be repeated for years, leading to brain deposition. Many applications of AI-powered contrast media are likely to develop in the next few years in the field of body imaging, as demonstrated by the growing evidence of successful malignancy characterization on non-contrast MR with the use of AI [118,119]. Specifically, the characterization of liver (for which there are already several studies) and prostate cancer may significantly benefit from these techniques, decreasing unessential exposure to contrast media. Another promising field of application is pediatric imaging. Contrast media toxicity and deposition is even more concerning in the pediatric population due to long life expectancy. AI-powered contrast media may find fertile ground in pediatric imaging, such as in the follow-up of demyelinating disorders [120], metabolic diseases [121,122], phacomatoses [123], and post-treatment changes [124]. Furthermore, AI architectures are subject to constant growth and improvement. The implementation of graphical interfaces (GUIs), through which it is possible to fully exploit the generative potential of GANs architectures [125], is currently under development. These advancements will lead to broader clinical applications of virtual and augmented contrasts in the near future.

## Figures and Tables

**Figure 1 pharmaceutics-14-02378-f001:**
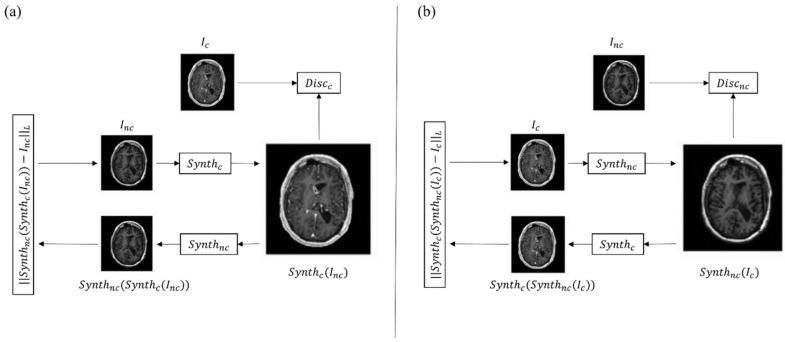
The CycleGAN model consists of a forward cycle and a backward cycle. (**a**) In the forward cycle, a synthesis network $Synth_{c}$ is trained to translate an input non-contrast image into a contrast one. Network $Synth_{nc}$ is trained to translate the resulting contrast image back into a non-contrast image that approximates the original non-contrast one. $Disc_{c}$ discriminates between real and synthesized contrast images. (**b**) In the backward cycle, $Synth_{nc}$ synthesizes non-contrast images from input contrast images, $Synth_{c}$ reconstructs the input contrast image from the synthesized non- contrast one, and $Disc_{nc}$ discriminates between real and synthesized non-contrast images. $I_{nc}$ = original non-contrast image; $I_{c}$ = original contrast image.

## Data Availability

Not applicable.

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
