# Peer review of "Synthetic Post-Contrast Imaging through Artificial Intelligence: Clinical Applications of Virtual and Augmented Contrast Media"

_pharmaceutics, 2022, doi:10.3390/pharmaceutics14112378_

Round 1

Reviewer 1 Report

The paper entitled "Synthetic Post-Contrast Imaging through Artificial  Intelligence: Clinical Applications of Virtual and Augmented Contrast Media" by Pasquini et al. is a review manuscript, which describes the current research and trends in application of artificial intelligence in the field of biomedical imaging. In general, the article is well organized and the structure is common among similar works. The flow of the text is good and the references are properly chosen for the topic of the manuscript. However, there are minor aspects, that could increase the readability and the overall outcome of the review.
1) In my opinion, there is lack of tables summarizing advances in discussed application fields. It seems necessary to summarize each chapter with a table where the most important information will be collected (data type, results, database, etc.)

2) The concept of developing the software, which is actually an application of previously developed model, is not new, the authors have mentioned some of the difficulties involved in this issue. The authors could discuss more in detail the danger of it (robustness, validation, etc.)?

3) If available, please provide public databases which are focused on the biomedical imaging?

Minor editorials:

Typos:

1) line 86 "groudtruth", should be "ground truth"

2) Abbreviations: line 371 "IV", should be "intravenous (IV)"

The above mentioned issues do not depreciate the overall good rating of the article, therefore I recommend the Editorial Office to proceed with the publication process after minor revision.

Author Response

10/25/22

Dear Editor and Referees,

We sincerely thank you for taking the time to review our work. We found the comments appropriate and helpful in improving the relevance of our research. We have addressed every comment in the paper and we have reported the answers to each query in this letter.

Reviewer 1

The paper entitled "Synthetic Post-Contrast Imaging through Artificial  Intelligence: Clinical Applications of Virtual and Augmented Contrast Media" by Pasquini et al. is a review manuscript, which describes the current research and trends in application of artificial intelligence in the field of biomedical imaging. In general, the article is well organized and the structure is common among similar works. The flow of the text is good and the references are properly chosen for the topic of the manuscript. However, there are minor aspects, that could increase the readability and the overall outcome of the review.

1) In my opinion, there is lack of tables summarizing advances in discussed application fields. It seems necessary to summarize each chapter with a table where the most important information will be collected (data type, results, database, etc.)

We sincerely thank the reviewer for reading our manuscript carefully and for pointing out the importance of adding summary tables to our chapters. We have summarized the main findings of every chapter in a dedicated table. We have highlighted the origin of the database used for training, specific artificial intelligence algorithms, tasks and type of acquisition. All these details are available in Table 1, Table 2 and Table 3.

2) The concept of developing the software, which is actually an application of previously developed model, is not new, the authors have mentioned some of the difficulties involved in this issue. The authors could discuss more in detail the danger of it (robustness, validation, etc.)?

We thank the reviewer for this suggestion. We expanded our discussion on the topic by adding a few remarks on the importance of model validation in the early stages of a study, as well as in the monitoring of model performance.

3) If available, please provide public databases which are focused on the biomedical imaging?

We thank the reviewer for giving us this input. We have included this information in Table 1, 2 and 3.

Minor editorials:

Typos:

1) line 86 "groudtruth", should be "ground truth"

We made the requested correction.

2) Abbreviations: line 371 "IV", should be "intravenous (IV)"

We made the requested correction.

The above mentioned issues do not depreciate the overall good rating of the article, therefore I recommend the Editorial Office to proceed with the publication process after minor revision.

Reviewer 2 Report

Overall Review

This is a comprehensive review on applying AI techniques on image processing for neuroradiology and body imaging clinical applications. It could be very helpful and informative for readers interested in this topic. This review article can be improved by having more clear messages and structure among works from articles cited, to give audience a better experience reading it. Detailed comments are as follows:

Comments:

  1. Line 65: Citation 15-17 are all self-citations. Is it possible for the authors to add works from other authors as well to make it comprehensive?
  2. Line 81: The authors mentioned two main counterparts “virtual” and “physical” - Maybe I missed that but could the “physical” part also be explained as well?
  3. Line 181: Image quality of figure 1 needs to be improved. Some arrows are not shown in the manuscript. Also the text is somehow aliased.
  4. Line 162: The authors need to elaborate/reference articles about what is MOS score. It is not clear just from the equation.
  5. Line 169 and line 177: Neither SSIM and PSNR are not clearly described — What are they used for? What are x and y? Why can PSNR be used to quantify reconstruction quality?
  6. Line 203: The more-than-one-page paragraph is very hard to follow for readers and lack clear logic flow. Consider add subsections or subtitles for each takeaway message the authors would like audience to follow.

Author Response

Line 65: Citation 15-17 are all self-citations. Is it possible for the authors to add works from other authors as well to make it comprehensive?

We thank the reviewer for this advice. We have included additional more comprehensive references, as suggested.

Line 81: The authors mentioned two main counterparts “virtual” and “physical” - Maybe I missed that but could the “physical” part also be explained as well?

We thank the reviewer for pointing out this aspect of our discussion. After reviewing the corresponding section, we deemed our previous incipit too general and we decided to modify the paragraph in order to better serve the topic of the review and to avoid off-topic information. The new paragraph can be seen in the revised manuscript, lines 81-84.

Line 181: Image quality of figure 1 needs to be improved. Some arrows are not shown in the manuscript. Also the text is somehow aliased.

We kindly invite the reviewer to check the correction. Thanks again for giving us this recommendation.

Line 162: The authors need to elaborate/reference articles about what is MOS score. It is not clear just from the equation.

The reviewer is right, and in view of this, a reference has been added regarding the defined metric so as to clarify any doubts for readers. The addition can be seen in the revised manuscript, lines 183-184.

Line 169 and line 177: Neither SSIM and PSNR are not clearly described — What are they used for? What are x and y? Why can PSNR be used to quantify reconstruction quality?

We thank again the reviewer for giving us the opportunity to expand this section. All the changes have been included in the revised manuscript, lines 190-211.

Line 203: The more-than-one-page paragraph is very hard to follow for readers and lack clear logic flow. Consider add subsections or subtitles for each takeaway message the authors would like audience to follow.

We agree with the reviewer, and we added the requested subsections to the manuscript. We kindly invite the reviewer to check the revised version of paragraph 4.

Reviewer 3 Report

The present review is very important and well organized. The significance of the information included is high with respect to the increasing necessity of contrast materials for medical applications. The idea of applying AI approaches for overcoming some problems with contrast media is very advanced and helpful.

I have some comments and suggestions to the authors:

1. Are CNN and GANs the only effective tools and algorithm of AI strategy for improving the medical imaging? This question needs some additional explanation in the manuscript body.

2. Is so, both approaches have to be explained in more details in the theoretical part, not only by respective references. This is done for the model validation section of the review.

Author Response

The present review is very important and well organized. The significance of the information included is high with respect to the increasing necessity of contrast materials for medical applications. The idea of applying AI approaches for overcoming some problems with contrast media is very advanced and helpful.

I have some comments and suggestions to the authors:

  1. Are CNN and GANs the only effective tools and algorithm of AI strategy for improving the medical imaging? This question needs some additional explanation in the manuscript body.

We sincerely thank the reviewer for giving us the opportunity of explaining this point. At the current stage, the tools mentioned in our manuscript (GAN and CNN) are the only ones available for deep-learning image reconstruction. We have emphasized this point in the manuscript (line 94-97), in accordance with prior published reports (1)

  1. If so, both approaches have to be explained in more details in the theoretical part, not only by respective references. This is done for the model validation section of the review.

As the reviewer kindly advised, we have expanded the manuscript to include additional details of the reported networks. The new content can be found in the lines 122-123 for CNN U-net based networks and in the lines 150-157 for CycleGANs.
